# Deficient Leptin Cellular Signaling Plays a Key Role in Brain Ultrastructural Remodeling in Obesity and Type 2 Diabetes Mellitus

**DOI:** 10.3390/ijms22115427

**Published:** 2021-05-21

**Authors:** Melvin R. Hayden, William A. Banks

**Affiliations:** 1Departments of Internal Medicine, Endocrinology Diabetes and Metabolism, Diabetes and Cardiovascular Disease Center, University of Missouri-Columbia School of Medicine, One Hospital Drive, Columbia, MO 65212, USA; mrh29pete@gmail.com; 2Geriatrics Research, Education and Clinical Center, Veterans Affairs Puget Sound Health Care System, 1660 S. Columbian Way, 810C/Bldg 1, Seattle, WA 98108, USA; 3Department of Medicine, Division of Gerontology and Geriatric Medicine, University of Washington, Seattle, WA 98108, USA

**Keywords:** adipose tissue, blood-brain barrier, blood-cerebrospinal fluid barrier, endothelial cell, endothelial glycocalyx, permeability, aging, insulin resistance, leptin resistance, neurovascular unit, obesity, microglia, ultrastructure

## Abstract

The triad of obesity, metabolic syndrome (MetS), Type 2 diabetes mellitus (T2DM) and advancing age are currently global societal problems that are expected to grow over the coming decades. This triad is associated with multiple end-organ complications of diabetic vasculopathy (maco-microvessel disease), neuropathy, retinopathy, nephropathy, cardiomyopathy, cognopathy encephalopathy and/or late-onset Alzheimer’s disease. Further, obesity, MetS, T2DM and their complications are associated with economical and individual family burdens. This review with original data focuses on the white adipose tissue-derived adipokine/hormone leptin and how its deficient signaling is associated with brain remodeling in hyperphagic, obese, or hyperglycemic female mice. Specifically, the ultrastructural remodeling of the capillary neurovascular unit, brain endothelial cells (BECs) and their endothelial glycocalyx (ecGCx), the blood-brain barrier (BBB), the ventricular ependymal cells, choroid plexus, blood-cerebrospinal fluid barrier (BCSFB), and tanycytes are examined in female mice with impaired leptin signaling from either dysfunction of the leptin receptor (DIO and *db/db* models) or the novel leptin deficiency (BTBR *ob/ob* model).

## 1. Introduction

Obesity may be considered a chronic disease and is a global problem [1]. Obesity is primarily driven by high caloric diets (Western or cafeteria diet) and sedentary lifestyles [2]. The Western diet and sedentary lifestyles have contributed to the aging baby-boom generation of our global society of obesity and Type 2 diabetes mellitus (T2DM). Additionally, the post-World War II baby boom generation is aging, a trend that is expected to continue over the next two to three decades (2020 to 2050). Importantly, we may be currently living in one of the oldest-living global populations with T2DM and late-onset Alzheimer’s disease (LOAD) each being age-related diseases [3].

In addition to diet and physical activity, genetic predisposition plays an important role in weight gain and obesity [4]. The genetically induced hyperphagia and obesity in *db/db* mice, *ob/ob* mice, BTBR.Cg-Lepob/WiscJ *ob/ob* (BTBR *ob/ob*) mice and Zucker obese *fa/fa* rats are associated with the development of insulin resistance, Type 2 diabetes mellitus (T2DM), and altered leptin signaling. In the human population, genetic variation contributes to the polygenic and multifactorial etiologies associated with obesity, metabolic syndrome (MetS), insulin resistance, and T2DM [3,4]. These, in turn, contribute to multiple end-organ aberrant remodeling and end-organ disease complications such as diabetic vasculopathy (macro-microvessel disease), neuropathy, retinopathy, nephropathy, cardiomyopathy, diabetic cognopathy, and late-onset Alzheimer’s disease (LOAD). In this review, the role of the adipocyte-derived hormone (adipokine) leptin in multiple end-organ diabetic-opathies and especially in aberrant ultrastructure remodeling of the brain will be emphasized. Early on, even prior to the clinical diagnosis of T2DM, there may be variable degrees of obesity, insulin and leptin resistance and impaired glucose tolerance. Therefore, one can model MetS or T2DM as a clinical spectrum disease that in combination with disease duration, aging, and other variables evolves over time [5].

Two position statements by the American Association of Clinical Endocrinologists (AACE) include important views on prediabetes or impaired glucose tolerance and overt T2DM as “dysglycemia-based chronic disease” and obesity or adiposity-based chronic disease. These statements include a more broad-based view of each of these two chronic diseases that are now considered of global importance regarding their pathophysiology and progression, such that they are often referred to as a common chronic disease of “diabesity” [6,7,8].

Preclinical rodent models have played an important role in studying the effects of impaired leptin signaling. Therefore, it is important to place in perspective a historical timeline of preclinical models used in research and development to study the effects of leptin in obesity and T2DM (Figure 1) [9,10,11,12,13,14,15,16,17,18,19]. Preclinical research and discovery take a considerable amount of time and patience as noted in the progression of discovery in the following timeline (Figure 1).

Occasionally, a marked development or discovery is made which changes the way we think. Some discoveries may even result in a paradigm shift in how we view a certain clinical disease state, such as that of leptin by Friedman and colleagues [13,14,15,20]. Figure 1 illustrates the historical timeline regarding animal models of obesity and T2DM, including the discovery of leptin. These models are central to understanding the role leptin plays in ultrastructural remodeling of the brain’s neurovascular unit, the capillary bed of the brain, which forms the blood-brain barrier (BBB), and the choroid plexus which forms the blood-cerebrospinal fluid barrier (BCSFB).

The BTBR *ob/ob* model described by Hudkins et al. is a recent model, mostly used to study diabetic nephropathy [19]. Briefly, the BTBR.Cg-Lepob/WiscJ *ob/ob* mouse (BTBR *ob/ob*) is characterized by early insulin resistance with elevated insulin levels, pancreatic islet hypertrophy, and the development of hyperglycemia by six-weeks of age. Crossing the BTBR strain with the *ob/ob* mutation results in the BTBR *ob/ob* model, characterized by diabetes with glucose levels in the range of 350 to 400 mg/dl. Unlike the *ob/ob* model, hyperglycemia in the BTBR *ob/ob* mouse is sustained and by 20-weeks of age, both sexes show similar levels of glucotoxicity. The BTBR *ob/ob* model, highly preferred for study of diabetic nephropathy, is largely unexplored regarding brain remodeling [19].

## 2. Leptin

Leptin (derived from the Greek *leptos* meaning “thin”) is an adipokine-polypeptide hormone with a molecular mass of 16kD consisting of 167 amino acids encoded by the obesity (ob) gene. It is primarily synthesized and secreted by the subcutaneous white adipose tissue (WAT) and the organ centric omental-visceral adipose tissue (VAT), which includes the perivascular adipose tissue (PVAT) or tunica adiposa adipocytes [13,14,15,21,22]. Interestingly, subcutaneous WAT produces more leptin than VAT and is now considered an endocrine tissue in addition to its earlier known role as a storage depot for excess energy intake [13,14,15,21,22]. Leptin expression is regulated by a variety of hormones, including insulin, glucocorticoids (corticosterone in rodents and cortisol in humans) and even leptin itself [21,22]. Circulating leptin levels are known to be in proportion to body adipose mass and thought to serve as an adiposity signal of total body energy stores to the brain hypothalamic nuclei [23,24]. Leptin is proposed to act as an afferent signal in the negative feedback loop to the hypothalamus that inhibits food-intake, controls energy homeostasis and thermogenesis, and regulates adipose tissue mass. Importantly, leptin is capable of autocrine (self), paracrine (adjacent) and endocrine signaling to distant tissues including the brain [14,15,16,17,25,26,27].

The concept of leptin resistance (LR) is important to the understanding of obesity in humans as well as in the diet-induced obesity (DIO) Western and *db/db* models. Indeed, hyperleptinemia is thought to result from LR resulting from deficient cellular signaling by leptin [27,28].

Gruzdeva et al. have suggested that various mechanisms may underlie LR in the brain and include a number of possible molecular and functional alterations, which may be characterized by structural changes to the leptin molecule, its transport across brain barriers, and leptin-receptor dysfunction/impaired signaling [28]. Deficient leptin cellular signaling, whether because of deficient leptin secretion, faulty leptin transport at the BBB and BCSFB interfaces, genetic abnormalities of leptin receptors, or post-leptin receptor signaling defects, results in loss of leptin’s neuroprotective effects and results in changes in brain function and remodeling [27,28,29].

The receptor for leptin (LepR or OB-R) is a Type I cytokine receptor protein encoded by the rather ubiquitous LEPR gene and is present in adipose tissue, brain, cardiovascular tissue, liver, kidney, skeletal muscle, and other tissues [30,31]. Splicing variants give rise to several forms of the leptin receptor, but it is the long isoform (LepRb) that participates in intracellular signaling. LepRb is highly expressed in the hypothalamus, where energy homeostasis and neuroendocrine function is regulated [31,32]. The short isoform of LepR has been proposed to mediate the transport of leptin across the blood-brain barrier, but currently there is evidence both for and against this proposal [33,34].

The leptin receptor is widely distributed and leptin has multiple pleiotropic effects [14,15,16,17,25,26,27]. As examples, leptin is important in the embryologic development of the brain and modulates glucose homeostasis, neuroendocrine axes, the autonomic nervous system, memory, and neural plasticity [35]. Leptin is capable of modulating multiple non-satiety processes, which include thermogenesis, reproduction, angiogenesis, osteogenesis, hematopoiesis, immune functions, cardio-cerebrovascular functions at the level of the myocardial capillaries and brain capillary endothelial NVUs, and renal glomeruli [19,20,21,22,23,24,25,26,27,28,29,30,31,32,33,34,35,36]. As discussed later in Section 4.1.2, these functions may importantly extend to regulation of the brain endothelial cell(s) (BEC) glycocalyx.

## 3. Central Nervous System (CNS) Roles of Leptin in Diet induced Obesity (DIO), *db/db* and BTBR *ob/ob*: Genetic Preclinical Models

Over the past few years, we have studied the effects of leptin on brain ultrastructural remodeling [3,37,38,39,40,41,42,43]. This work is reviewed in the following sections. In obese rodent models (including the DIO-Western, *db/db*, and BTBR *ob/ob*), the effects of leptin signaling in the brain and peripheral tissues has been studied. Herein, we also include the streptozotocin (STZ) induced diabetes model (STZ-induced DM) in which animals have little adipose tissue, insulin, or leptin. It is important to note that to date, the only published work in this field on the BTBR *ob/ob* model is a poster presentation at the 2019 ADA poster presentations [44]; however, in this review, we will illustrate some ultrastructural remodeling changes, noting that the BTBR *ob/ob* mouse has an increased permeability (i.e., leakiness of the BBB) in most regions of the brain (Figure 2A,B and Figure 3) [44]. Figure 2 compares three obesity models with dysglycemia (DIO, *db/db*, BTBR *ob/ob*) to their non-diabetic controls and to the STZ-induced model of T1DM.

## 4. Neurovascular Unit (NVU) as an Anatomical Ultrastructural and Functional Unit

The NVU includes the brain endothelial cells which form the blood-brain barrier (BBB), basement membranes (BM), pericytes (Pc) vascular smooth muscle cells in arterioles, microglia, astrocytes (AC), and neurons (myelinated and/or unmyelinated). The cell types forming the NVU are in cross-talk with one another, influencing one another’s behavior. Oligodendrocytes and myelinated and unmyelinated neuronal axons can exist in the BM, especially in the subcortical and white matter regions of the brain (Figure 4). Various cells within the NVU are responsible for BBB formation and autoregulation of blood oxygen level dependent (*BOLD*) regional cerebral blood flow (CBF), measurable by positron emission tomography and functional magnetic resonance imaging [3,37,38,39,40,41,42,43]. 

### 4.1. The Brain Endothelial Cell (BEC)

The elongated squamous-like monolayer of brain endothelial cells of the NVU defines the luminal surface of the NVU. The peripheral vascular endothelial cells have cell-cell junctions consisting of vascular endothelial cellular adhesion molecules or cadherins (VE-cadherins); however, the CNS is unique in that BECs also have tight junctions consisting of occludins, claudins, and junctional adhesion molecules (JAMs) in addition to VE-cadherins. The combination of the protective tight and adherens junction (TJ/AJ) proteins allow for decreased permeability in the brain and are critical for the formation of the BBB, BCSFB, and the barrier formed by arachnoid vascular endothelial cells [45,46]. Additionally, the tanycytes that line the ventricles in the region of median eminence (ME) of the hypothalamus and other circumventricular organs (CVOs) have tight junctions between non-ciliated epithelial cells, mediating the entry of leptin to the hypothalamic nuclei from the cerebrospinal fluid (CSF) [47,48]. Additionally, the BECs provide not only a specialized extracellular matrix (ECM) at the luminal surface termed the endothelial glycocalyx (ecGCx), but also at its abluminal surface where the ECM is termed the basilar lamina or basement membrane (BM).

#### 4.1.1. The BECs and Endothelial Glycocalyx (ecGCx)

The intact ecGCx is important for the vascular integrity of arteries, arterioles, and capillaries. The ecGCx is composed of a sugar-protein mesh, gel-like slime surface coating the luminal apical polarized BEC monolayer (Figure 5). This protective coating modulates the direct contact of blood and its constituents with the plasma membrane of the BEC. The ecGCx is primarily synthesized by the BECs with some contribution by plasma albumin, orosomucoids, fibrinogen, glycoproteins, and glycolipids [49,50,51,52,53]. The ecGCx is anchored to the BEC luminal plasma membranes by highly sulfated proteoglycans (syndecans and glycipans), glycoproteins (including selectins such as various cellular adhesion molecules and integrins) and non-sulfated hyaluronan (a glycosaminoglycan) via BEC CD44 (Figure 5). Hyaluronan may also be free floating (unbound), attached to the assembly proteins such as the BEC hyaluronan synthases, or form hyaluronan-hyaluronan stable complexes [50,51,52,53]. It is very interesting that the ecGCx is very similar in its composition to the brain’s interstitial extracellular matrix (ECM) [53,54].

The ecGCx is anchored via the proteoglycan (glypican) to the caveolae and this plays a key role in mechanotransduction [55,56]. The ecGCx is important to both the overall barrier protection of the BEC and to the mechanotransduction of endothelial fluid shear stress. Luminal shear stress induces production of BEC-derived nitric oxide via the lipid rafts of the caveolae and the glycoproteins of the ecGCx to produce fluid shear stress-induced nitric oxide (NO) (Figure 6). This NO production is important for vasodilation of the aorta, the carotid and vertebral arteries, the choroid (anterior and posterior) arterioles via the vascular smooth muscle cells and eventually to the pericytes of the pial arterioles, the capillary system to the fenestrated capillaries, and the choroidal microvessels flowing into the choroid plexus as discussed in Section 6 (Figure 14). If this mechanism is lost via a disturbance, attenuation, loss or shedding of the ecGCx, then there will be a decrease in bioavailable NO to signal vascular smooth muscle in cerebral arterioles or pericytes in capillary neurovascular units in the brain. The attenuation or loss of bioavailable NO could play an important role in providing proper cerebral blood flow and neurovascular coupling and may have highly detrimental effects resulting in brain pathology, such as dysfunctional synapses and/or loss of synapses and neurodegeneration.

The ecGCx is extremely fragile and is in a constant state of flux (synthesis-loss being regenerated and repaired by the BECs and from plasma constituents) and if one of the components is attenuated or lost then the entire ecGCx may result in a loss of function and/or collapse or be shed. Importantly, the ecCGx has a net negative charge due to heavy sulfation of the GAGs of PGNs, GPs and orosomucoids. This net negative charge allows for strongly positive stains such as lanthanum nitrate and the original ruthenium red with strong positive charge to stain the ecGCx via electrostatic (charge-) selective interactions.

Here, we investigated the ecCGx in the brains of control heterozygote littermates BTBR*ob*+/− and of obese, diabetic homozygous BTBR *ob/ob* models by lanthanum perfusion fixation (2% glutaraldehyde, 2% sucrose, 0.1 M sodium cacodylate buffer (pH 7.3) 2% lanthanum nitrate) prior to sacrifice as described by Ando et al. [57] and Okada et al. [58]. Whole brains were then placed in standard electron microscopic fixative (2% paraformaldehyde and 2% glutaraldehyde in 100 mM sodium cacodylate buffer, pH = 7.35) as previously described by Hayden et al. [39] with an n = 3 of each model at 20 weeks of age. Approximately 1 mm sections from the mid-cortical gray matter tissue and hippocampus (hippocampus CA-1) regions from left hemisphere were then rinsed with 100 mM sodium cacodylate buffer (pH 7.35) containing 130 mM sucrose. Secondary fixation was performed using 1% osmium tetroxide (Ted Pella, Inc., Redding, CA, USA) in cacodylate buffer using a Pelco Biowave (Ted Pella) operated at 100 W for 1 min. Specimens were next incubated at 4 °C for 1 h, then rinsed with cacodylate buffer, and further rinsed with distilled water. En bloc staining was performed using 1% aqueous uranyl acetate and incubated at 4 °C overnight, then rinsed with distilled water. Using the Pelco Biowave, a graded dehydration series (e.g., 100 W for 40 s) was performed using ethanol, transitioned into acetone, and dehydrated tissues were then infiltrated with Epon resin (250 W for 3 min) and polymerized at 60 °C overnight. Ultrathin sections were cut to a thickness of 85 nm using an ultramicrotome (Ultracut UCT, Leica Microsystems, Wetzlar, Germany) and stained using Sato’s triple lead solution stain and 5% aqueous uranyl acetate. Multiple images were acquired for study at various magnifications with a JOEL 1400-EX TEM JEOL (JEOL, Peabody, MA, USA) at 80 kV on a Gatan Ultrascan 1000 CCD (Gatan, Inc., Pleasanton, CA, USA). This method allows for consistent and reproducible observations of the ecGCx by transmission electron microscopy (TEM).

We [3] and others [57,58] have demonstrated the presence of an ecGCx in the wild type control models using lanthanum perfusion fixation staining to demonstrate an attenuation and/or loss of the ecGCx in the obese diabetic homozygous BTBR *ob/ob* models as compared to the heterozygous non-obese, non-diabetic control models in Layer III of the cortex (Figure 7) and in hippocampus (Figure 8). To date we have not studied the ecGCx in the DIO Western, obese diabetic *db/db*, or the streptozotocin induced Type I diabetic models [3,38,39,40,41,42,43,44].

We treated the homozygous obese diabetic BTBR *ob/ob* for 16-weeks prior to sacrifice with leptin (per implanted subcutaneous pumps at a dose of 15 µg/day). We found that the treatment with leptin protected the ecGCx from being attenuated (Figure 9).

The TEM images presented in this review demonstrate that the ecGCx is attenuated in the obese diabetic BTBR *ob/ob* and that leptin replacement ameliorated this attenuation. Images in control animals (heterozygotes) are very similar to the ones previously published by Ando et al. [57] and Okada et al. [58] who also used lanthanum nitrate staining to visualize the ecGCx. While these studies are not quantitated, they do provide evidence for future studies to access the ecGCx in obese/diabetic models.

The ecGCx is in a near continuous state of flux, being constantly shed and regenerated. Studies of the elusive ecGCx have recently gained increasing interest and it is thought that this interest will continue to increase as it becomes more reliably identified through multiple techniques, including the use of lanthanum nitrate staining for TEM identification. Recent studies suggest that the ecGCx is important to the integrity of NVU and BBB permeability. Kutuzov et al. have demonstrated in an elegant experiment that the ecGCx may be the first of three major barriers forming a tripartite BBB with the BEC forming the middle barrier and the abluminal BM, pericytes and Pcfp BM, and astrocytic end-feet forming the third barrier [59]. Additional in vivo techniques for studying the ecGCx should further add to our knowledge of this structure [51,60].

#### 4.1.2. The BEC Basement Membrane

The BEC basal lamina, commonly referred to as the BM, consists of Type IV collagen, fibronectin, laminin, nidogen, and heparan sulfate proteoglycans (agrin and perlecan). It envelops the basilar portion of the BEC, splits to encompass the pericyte (Pc) and forms the anchoring structure for the corona of astrocyte foot processes (ACfps) (Figure 5) [3,40,43,54]. The BM is synthesized primarily by the EC with both pericytes and astrocytes contributing to its synthesis. Interestingly, the polarized BECs create barriers on the luminal surface layer (ecGCx), the paracellular sides by forming the TJ/AJ, and the abluminal BM. In diabetic *db/db* models, there is BM thickening and even though it is thickened, its structure is remodeled, allowing for increased permeability [54]. Interestingly, BM thickening was not observed in the STZ-DM, DIO, or the BTBR *ob/ob* models we have studied.

#### 4.1.3. BEC Activation-Dysfunction of the NVU

BEC activation-dysfunction is one of the prominent findings in obesity, MetS, and T2DM and contributes to the dysfunction of the brain NVUs with impaired CBF and NVU coupling with neurons. BEC activation and dysfunction have been visualized by TEM in the *db/db* obese diabetic model with leukocytes, red blood cells and platelets adhering to the activated BEC and is associated with activated microglia cells (aMGCs) [3,40,41]. In the BTBR *ob/ob* models in both cortical and hippocampal regions of the brain, we observed aMGCs encompassing BECs and encroaching the NVU space in the CA-1 regions of the hippocampus (Figure 10).

### 4.2. The Neurovascular Unit Pericyte (Pc)

In the obese diabetic *db/db* model, Pcs are decreased in number and have aberrant mitochondria and retracted nuclei [3,40]. Pericytes were also retracted or lost in streptozotocin-induced DM [39]. The Pcs are intact in all of the NVUs of the BTBR *ob/ob* except when there was an encompassing aMGC as previously depicted (Figure 10C,D).

### 4.3. Neurovascular Unit Astrocyte (AC) and Astrocyte Foot Process (ACfp)

ACfp are detached and retracted in the *db/db* [3,40,44] and AC plasma membrane ruffling is present in the streptozotocin-induced DM model [39]. The BTBR *ob/ob* models revealed intact ACs in the cortex, hippocampus, and in the midbrain regions except when there was an encroaching aMGC as depicted in the hippocampus (Figure 10C,D).

### 4.4. NVU Oligodendrocytes and Myelinated Neurons

Oligodendrocyte and neuronal axon myelin were markedly remodeled in the obese diabetic *db/db* models [42]. In comparison, there were no abnormalities of oligodendrocyte or myelin detected in the current study of this female BTBR *ob/ob* model. The *db/db* models demonstrated marked dysmyelination with outer myelin lamellae sheath splitting, separation and ballooning with aberrant mitochondria in grey matter and subcortical regions with axonal collapse in both the transitional and subcortical layers. Further, there were prominent remodeling changes of the oligodendrocytes with increased volume of nuclear chromatin condensation.

## 5. Microglia (MGC)

As noted above, aMGC encroaching into the NVU and encompassing the BEC are a frequent finding in the BTBR *ob/ob* model in the hippocampus. In the cortex, ramified microglia were increased compared to controls, but there was no activation in cortical Layer III of microglia. Ramified microglial cells (rMGCs) protect brain functions by phagocytosing cellular debris. MGCs are of mesodermal (yolk sac derived) origin and serve the brain as resident innate immune cells. They are responsive to many cytokines, chemokines, and signaling molecules, producing free radicals (superoxide, reduced nicotinamide adenine dinucleotide phosphate (NADPH Ox), inducible nitric oxide and mitochondrial-derived reactive oxygen/nitrogen (mtROS). The aMGCs are able to return to their surveilling-ramified phenotype once the invaders or danger–damage signals have been eradicated. Additionally, MGCs are important in brain development and play an important role shaping and pruning neuronal synaptic connectivity in formative and adult years [41,61,62,63,64,65,66,67,68,69,70,71]. MGCs are distinct from bone marrow derived peripheral monocyte-macrophage cells in that they are the brain resident inflammatory immune cells and not dependent on recruitment from the peripheral systemic circulation but are capable of undergoing proliferation and activation as needed [72]. Multiple toxicities in obesity and T2DM including the absence of leptin cellular signaling may result in aMGCs in the *db/db* models to a much greater extent as compared to the BTBR *ob/ob* models (Figure 11) [3,41,43].

We observed an increase in the number of rMGCs in the cortical Layer III of the BTBR *ob/ob* as compared to control models and this may be due to an increase in metabolic neuronal and NVU activity in these regions. In the hippocampal regions of the BTBR *ob/ob*, we not only observed a marked increase in rMGCs in excess of the cortical regions as compared to control models, but also noted the encroaching aMGCs (Figure 10 and Figure 11).

Our results in the BTBR *ob/ob* model strongly suggest that these models are a pro-neuroinflammatory phenotype, especially in the hippocampal regions. Interestingly, the difference between the level of activation of the microglia in the *db/db* and BTBR *ob/ob* suggests a difference in their genetic background even though both models were obese and developed T2DM with hyperglycemia at 20-weeks. In the original publication describing the BTBR *ob/ob* model, Hudkins et al. [19] noted that glucose levels were in the 300 mg/dL range and that in the *db/db* the mean glucose at a similar age were in the range of 500 mg/dL [3,41,43,73,74]. This difference in the amount of glucotoxicity could be at least one reason the activation of MGC were not as prevalent in the female BTBR *ob/ob* as compared to the *db/db* models [19]. Additionally, BTBR *ob/ob* fetuses, unlike *db/db* fetuses, will have leptin stimulation probably until weaning. Ultrastructural observations in the *db/db* and BTBR *ob/ob* certainly implicate the activation of resident microglia and encroachment of the NVU by aMGCs may be one of the mechanisms for impaired NVU capillary dilation due to regional neuron signal or NVU uncoupling associated with decreased regional cerebral blood flow within the hypothalamus [3,41]. This NVU uncoupling could certainly be related to a decrease in CBF and neuronal ischemia resulting in synaptic dysfunction and/or neuronal loss with ensuing impaired cognition and regional brain atrophy. However, brain atrophy has only been reported in the *db/db* models by Ramos-Rodrigues et al. [75] and was not found to be present in this current BTBR *ob/ob* study (Figure 12).

Ramos-Rodriguez et al. [75] have demonstrated that the *db/db* models have abnormal gross morphological findings with decreased brain weights, including grossly evident cortical parietal-hippocampal atrophy, a finding not present in the BTBR *ob/ob* model (Figure 12). These findings suggest an interaction of genetic background with obesity and T2DM in regards to brain morphology as well as the role of leptin receptor deficiency vs deficient leptin production as in the BTBR *ob/ob* models. Whether this atrophy relates to differences in neurogenesis, decreased CBF, or neuronal ischemia with loss of synapses and neurons is unknown.

## 6. Blood-Cerebrospinal Fluid Barrier (BCSFB), Choroid Plexus, Median Eminence of Third Ventricle, Tanycytes, Circumventricular organ(s) (CVOs) and Hypothalamic Nuclei

The brain’s ventricular system (two lateral and third, aqueduct, and fourth ventricles) contain the cerebrospinal fluid (CSF) and are lined primarily by cuboidal epithelial cells termed ependymal (EPY) cells (Figure 13) [76].

EPYs have CSF apical microvilli that increase their surface area and cilia to aid the movement of CSF within the ventricles. EPYs have a genetic linkage to the neuroepithelial cells of the developing neural tube of the embryo. CSF within the ventricles is thought to serve as a sink for the deposition of toxic metabolic byproducts created by the high metabolic neuronal activity within the brain and their subsequent removal into the systemic venous pathway (including the toxic amyloid beta proteins). The CSF from the ventricles enters the subarachnoid space to be drained by the subarachnoid granulations and nasal lymphatics into the systemic circulation [77,78,79,80,81].

The rather uniform coverage of the ventricles by EPYs gives way to a specialized covering of the choroid plexus (CP) in the floor of the lateral ventricles and the roof of the third ventricle. The CP is supplied by the choroid arterial system and these end in a plexus of capillaries that are fenestrated. These fenestrated capillaries produce a plasma filtrate that is separated from the CSF by the EPYs that form the CP. Importantly, these EPYs of the CP possess TJ/AJ and secrete the bulk of CSF (Figure 14) [48,82].

The CP is primarily responsible for two-thirds of the CSF production along with the EPY cells and cells lining the subarachnoid space [83]. Of significant importance is the recent report that insulin is synthesized in the EPY cells of the choroid plexus, which is regulated by serotonergic signaling [84]. The CP regions are covered by the EPY cells as previously imaged and each of these adjacent EPY cells are known to contain TJ/AJ, forming the barrier function of the CP and the BCSFB (Figure 13 and Figure 14).

As one traces the ciliated EPY cells to the median eminence of the hypothalamus, the EPY cells give way to tanycytes, which are highly specialized ependymal cells, and lose their cilia, supposedly to better sense the multiple hormonal and non-hormonal proteins and polypeptide contents in the CSF. Importantly, the tanycytes also have elongated subventricular zone (SVZ) cytoplasmic processes that may directly interact with the portal fenestrated capillaries of the CVOs and are also capable of connecting the median with the hypothalamus.

Tanycytes were originally documented and named by Horstmann in 1954 [85]. Further, tanycytes have been discussed in detail by Rodriguez et al. [86] with more recent discussions provided by Gao et al. and Balland et al. [87,88], as well as in regards to the neuroimmune axis of the barriers [88,89]. Currently there are thought to be at least four types of tanycytes (alpha1, 2 and beta 1, 2-dorsal β1d, ventral β1v, lateral β2la and medial β2me) which have varied functions. The tanycytes are not only responsible for glucose sensing via their glucose transporters but also leptin sensing via its leptin receptors (LepRs) [89,90]. As one follows the lining EPYs of the CSF ventricles to the region of the median eminence the lining EPYs differentiate to become tanycytes, which are one mechanism for the uptake and distribution of leptin to the hypothalamus.

Intraventricular CSF lining tanycytes are highly specialized bipolar ependymal cells that line the ventrolateral wall and the floor of the third ventricle and play a significant role in leptin uptake at the level of the median eminence of the hypothalamus. Tanycytes also play a key role in our obesity and T2DM models in this review (DIO Western, *db/db* and BTBR *ob/ob*) as they sense leptin and glucose [91,92]. Therefore, tanycytes play a necessary and essential role in regulating peripheral metabolic signals and energy balance since leptin is considered to be the afferent arm of the brain’s negative feedback control of this delicate balance between peripheral WAT adiposity and the CNS response [90,91]. The above dual vascular supply of leptin via the pial BBB capillaries and the choroid capillaries of the CP; BCSFB are very important for leptin brain hypothalamic signaling and maintaining leptin CNS homeostasis [48,93].

In the BTBR *ob/ob* females, we found evidence of remodeling of the ventricular system (Figure 14). In particular, the basilar infoldings were highly vacuolated and vesiculated. The presence of microthrombosis illustrates the risk of vascular to the CP as well as to brain parenchyma.

## 7. Summary

When viewing adipose tissue as an endocrine tissue responsible for the production of the adipokine/hormone leptin, deficient leptin cellular signaling in either hyperleptinemia/leptin resistance models (DIO and the *db/db* mice) or a hypoleptinemia model (BTBR *ob/ob* mice) resulted in ultrastructural brain remodeling. In this review, we have examined brain ultrastructural remodeling in females of these three models: (i) a DIO model with elevated leptin and leptin receptor resistance with IGT-prediabetes; (ii) a *db/db* model with elevated leptin and leptin receptor resistance with obesity and T2DM; and (iii) the novel BTBR *ob/ob* without measurable leptin and with obesity and T2DM. We have demonstrated in these models the key role that deficient leptin cellular signaling plays in regards to ultrastructural brain remodeling. We further show that replacement of leptin may reverse the ultrastructural remodeling in the leptin deficient BTBR *ob/ob* mouse.

From this review, one can observe that deficient cellular signaling associated with DIO, *db/db*, and BTBR *ob/ob* is very complicated and some of these signaling abnormalities have yet to be completely understood, for example there may be other mechanisms such as tanycyte-independent control of hypothalamic signaling [94]. In addition, leptin is a multifunctional hormone, and therefore, when it affects one cell, those effects may not be able to be integrated with the particular cell and its functions that are being investigated in a study of impaired leptin cellular signaling especially in the brain [95].

Herein, we studied the effects of deficient leptin cellular signaling in the novel female model (BTBR *ob/ob*) at 20-weeks of age. Interestingly, we found that there were not any persistent significant remodeling changes to the EC TJ/AJs, pericytes or astrocytes of the NVU BBB in the BTBR *ob/ob* as we had previously found in the *db/db* models. Although there was endothelial cell activation due to BEC ecGCx attenuation in the BTBR *ob/ob* mouse, there was no markedly aberrant TJ/AJ, Pc, and AC remodeling as seen in the *db/db* mouse. Since there is preliminary evidence of increased BBB permeability inBTBR *ob/ob* [44], we decided to examine the ecGCx with lanthanum nitrate perfusion fixation and found observational evidence by TEM of an attenuation and/or loss of its BEC ecCGx, supporting the concept of a tripartite BBB. This finding suggests that attenuation/loss of the protective BEC ecGCx layer may contribute to the observed increase in permeability of the BBB in the leptin deficient mutation, obese, insulin resistant and diabetic BTBR *ob/ob*. The female BTBR *ob/ob* mice also demonstrated an ultrastructural phenotype of neuroinflammation with aberrant and activated microglia, especially in the hippocampal regions and aberrant remodeling of the CP basilar ependymal cells. Specifically, we observed novel aberrant vacuolization of the basilar infoldings of the ependymal cells of the choroid plexus in the lateral ventricles of the BTBR *ob/ob* model. This could result in CP dysfunction at the CP-CSF interface in obesity and T2DM with a deficiency of leptin cellular signaling.

Each of the models used in this review had different genetic backgrounds. The female DIO Western diet models (C57BL/6J) had free access to a high fat and sucrose-fructose and demonstrated obesity, insulin and leptin resistance, dysglycemia in the 150 to 200 mg/dL. This model can be considered prediabetic or impaired glucose tolerance model with insulin resistance at 16–20-weeks of age. The female *db/db* models consisted of the deficient LepR with markedly elevated levels of leptin, insulin and leptin resistance, obesity and diabetes with levels of glucose around 500 mg/dL. The female BTBR *ob/ob* models with genetic deficiency of leptin have unmeasurable leptin levels, insulin resistance, obesity, and diabetes with glucose levels in the 300 mg/dL range. The female mice in each of these models were sacrificed at 20–26-weeks of age. In the BTBR *ob/ob* models, we found less remodeling changes to the BECs NVU, but novel aberrant ecGCx remodeling changes response to leptin replacement.

Certain limitations apply to the comparisons made in this study. First, as with any TEM study there are certain limitations as a result of examining tissue at higher magnifications and at a single point in time. Here, we found the noted changes in multiple images. Second, the staining with lanthanum nitrate is novel; however, other studies in addition to ours have also demonstrated that this staining is reliable and reproducible. Third, this study was not correlated with functional data other than the increased permeability previously demonstrated [44]. Fourth, our studies were limited to female mice and some investigators have had previous concerns about studying female models; however, our ecGCx observations were similar to the ecGCx positive lanthanum nitrate staining findings in male mice by Ando et al. [57]. In conclusion, leptin, which has many behavioral and physiologic effects including on energy expenditure, thermogenesis, glucose and lipid metabolism, and sympathetic and parasympathetic [14,15] additionally has protective effects in the BTBR *ob/ob* mouse.

## Figures and Tables

**Figure 1 ijms-22-05427-f001:**
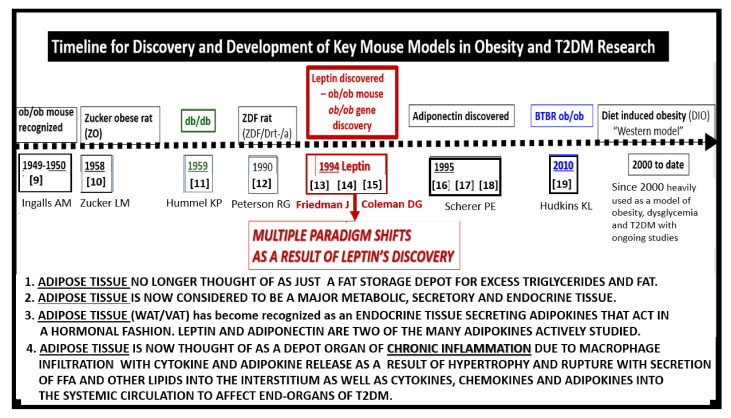
Timeline for discovery and development of key mouse models in obesity and Type 2 diabetes mellitus (T2DM) research. The dashed timeline illustrates important discoveries and models from 1949 to date. References are inserted below the dates of each model. The central importance of leptin is emphasized. FFA = free fatty acids; VAT = visceral adipose tissue; WAT = white adipose tissue.

**Figure 2 ijms-22-05427-f002:**
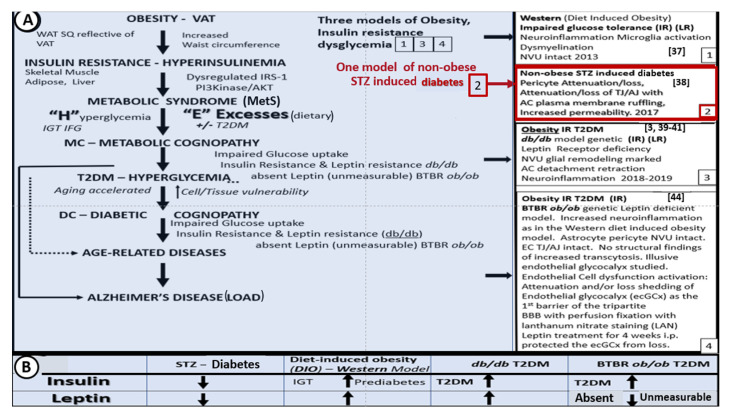
Brain ultrastructural remodeling in obesity models studied to date with insulin and leptin levels for comparison. **Panel A** depicts the importance of obesity, insulin resistance-deficient leptin cellular signaling in regards to the development of age-related diseases such as late onset Alzheimer’s disease (LOAD) and neurodegeneration (left-hand side blue coloration). Boxes 1, 3, and 4 (right-hand side) all share deficient cellular leptin signaling. Box 2 depicts the streptozotocin induced insulinopenic diabetes mellitus (DM). **Panel B** illustrates the increased (upward arrows) and decreased (downward arrows) of insulin and leptin in each model discussed in **Panel A**. AC = astrocyte; AJ = adherens junction; AKT = protein kinase B; EC = endothelial cell; IFG = impaired fasting glucose; IGT = impaired glucose tolerance; i.p. = intraperitoneal; IRS−1 = insulin receptor substrate−1; LAN = lanthanum nitrate; NVU = neurovascular unit; PI3Kinase = phosphoinositide 3-kinase; T2DM = Type 2 diabetes mellitus; SQ = subcutaneous fat; TJ = tight junction; VAT = visceral adipose tissue; WAT = white adipose tissue.

**Figure 3 ijms-22-05427-f003:**
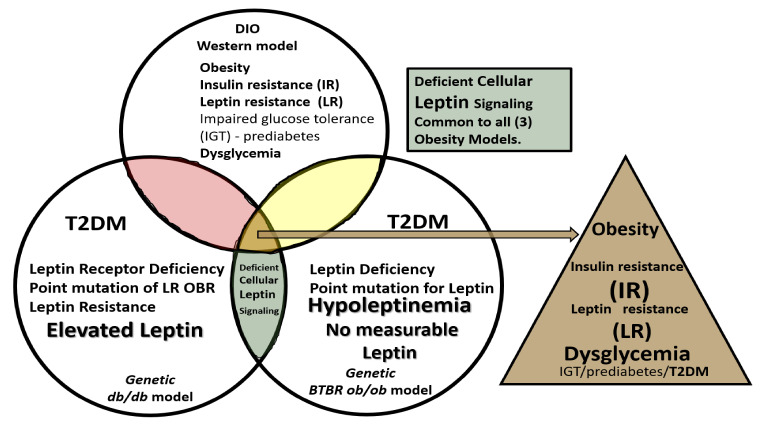
Venn diagram illustrates the shared importance of leptin in diet-induced obesity (DIO), Western, *db/db* and BTBR *ob/ob* models. Deficient cellular leptin signaling is common in all three models, but through different mechanisms. All models are obese and there is dysglycemia in the diet induced obesity (DIO) model, overt T2DM in the *db/db*, and elevated blood glucose levels in the BTBR *ob/ob*. IGT = impaired glucose tolerance; IR = insulin resistance; LR = leptin resistance; T2DM = Type 2 diabetes mellitus.

**Figure 4 ijms-22-05427-f004:**
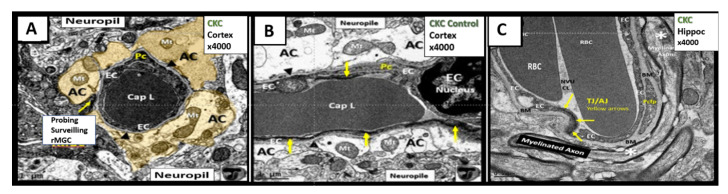
The neurovascular unit (NVU) in healthy control C57BL/6J mice at 20 weeks of age from cortical Layer III. **Panel A** astrocyte foot processes (AC) are pseudo-colored golden to emphasize their important anatomical location. Note the ramified microglial cell (rMGC) (yellow arrow) surveilling the NVU in **Panel A**. **Panel B** illustrates a horizontal image of the NVU (contrasting with the cross-section image in **Panel A**) to better illustrate the electron dense tight and adherens junctions (TJ/AJ) (yellow arrows) within the paracellular regions and emphasize the clear zone or corona of electron lucent AC foot processes (ACfp). **Panel C** depicts a NVU within the hippocampus where a myelinated axon bundle is present. Additionally, oligodendrocytes may exist in the NVU basement membrane (BM) especially in the subcortical and white matter regions (not shown). Note that the BM splits to encompass the pericyte foot processes (Pcfp). Magnification ×4000; scale bar = 1 µm. **Panels A** and **B** are adapted with permission from reference [39]. AC = astrocyte; ACfp = astrocyte foot processes; BM = basement membrane; Cap L and CL = capillary Lumen of the NVU; CKC = control C57BL/6J model; EC = endothelial cell; Mt = mitochondria; rMGC = ramified MGC; Pc = pericyte; Pcfp = pericyte foot processes.

**Figure 5 ijms-22-05427-f005:**
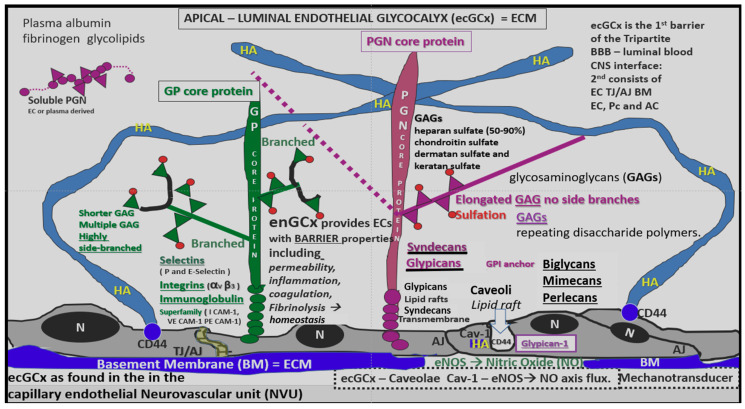
The endothelial glycocalyx as the first of component of the “tripartite BBB”. Normal components of the endothelial glycocalyx (ecGCx): a unique extracellular matrix. The ecGCx is composed of two classes of proteins that are mostly anchored [proteoglycan(s) (PGN) (purple), glycoprotein(s) (GP) (green)] and of hyaluronic acid (HA) hyaluronan (an exceedingly long polymer of disaccharides non-sulfated glycosaminoglycan). (HA) (blue) may be either unattached (free floating), attached to CD44 brain endothelial cell(s) (BEC) plasma membrane, or form HA-HA complexes. Non-sulfated HAs not anchored to the BECs may reversibly interact at the lumen with plasma-derived albumin, fibrinogen and soluble PGNs. The PGNs and glycoproteins side chains consisting of glycosaminoglycans (GAGs) covalently bound to core proteins are highly sulfated (red dots). The two primary PGNs are the syndecans and glypicans. The glycoproteins consist primarily of selectins (P and E), integrins (alpha v and beta 3) and immunoglobulin superfamily of ICAM-1, VE-CAM and PE CAM-1. Caveolae are invaginations of lipid rafts of the endothelial plasma membrane and contain CD44 important which anchors hyaluronan to the BEC plasma membrane and glycosylphosphatidylinositol (GPI) which anchors glypican-1. The GPI/glypican-1 interaction is thought to activate endothelial nitric oxide synthase (eNOS) to produce bioavailable nitric oxide (NO) via the calcium calmodulin dependent *Caveolin-1* (Cav-1) protein.

**Figure 6 ijms-22-05427-f006:**
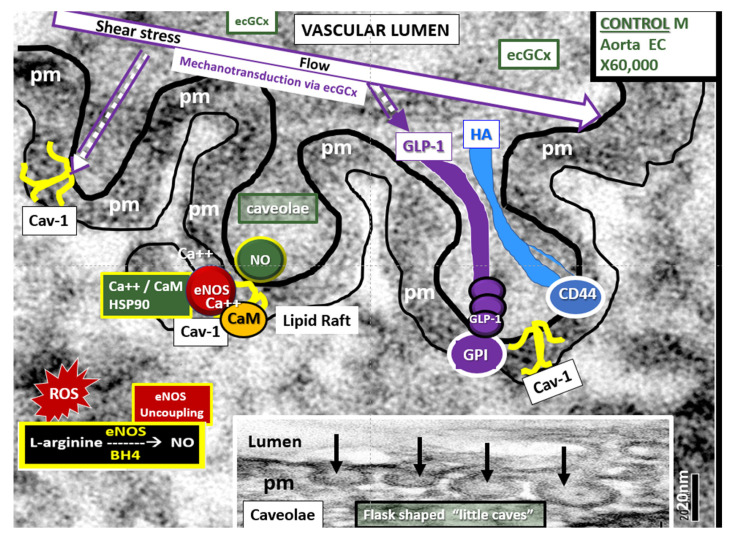
The properties of the endothelial cell glycocalyx (ecCGx) shield and mechanotransduction. These images of the aorta EC illustrate relations among the caveolae, the plasma membrane (pm, lined for clarity), and the ecGCx, with its glypican-1 (GLP-I) anchored to the lipid rafts of the caveolae. Note that the ecCGx extends into the caveolae. The Cav-1 protein is responsible for the omega shape (Ω) of the caveolae (yellow icon) (dashed arrow). The shear stress of laminar flow through the ecGCx induces mechanotranduction and release of NO, resulting in vasodilation (open arrow). The ecGCx component glypican-1 is the primary mechano-sensor for shear stress-induced NO production. Scale bar on the left panel is 20 nm and the right-hand panel is an exploded view of left panel. BH4 = tetrahydrobiopterin; Ca++/CaM = calcium calmodulin; Cav-1 = caveolin-1; GIP = glycosylphosphatidylinositol; GLP-1 = glypican-1; pm = plasma membrane.

**Figure 7 ijms-22-05427-f007:**
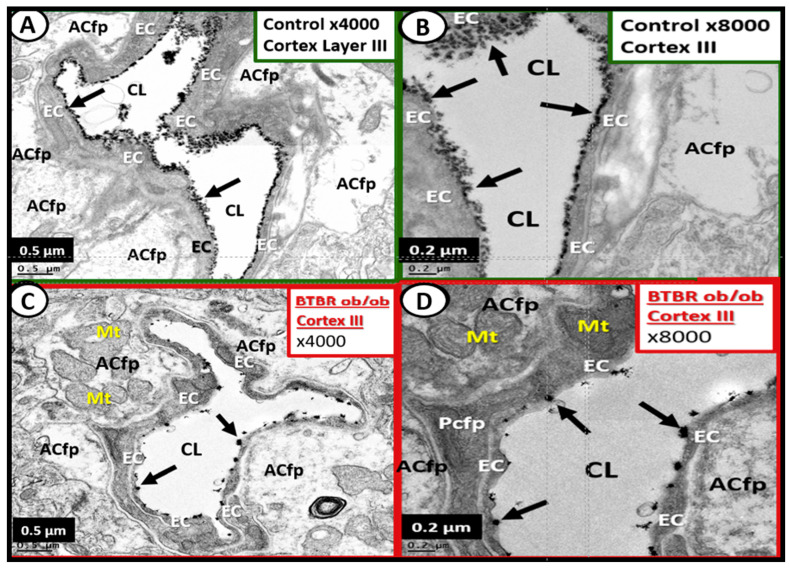
Lanthanum nitrate (LaN) staining of the endothelial glycocalyx (ecGCx) showing attenuation and loss in the diabetic obese BTBR *ob/ob* model: cortical Layer III at 20 weeks of age. **Panels A** and **B** demonstrate in the wild type control an intense electron dense LaN staining reflecting an intact continuous ecGCx or endothelial surface layer (arrows) of the brain endothelial cell(s) (BEC) at ×4000, ×8000; scale bars = 0.5, 0.2 µm respectively. In addition, note the intact astrocyte foot processes (ACfp), pericyte foot processes (Pcfp), and intact mito-chondria (Mt). **Panels C** and **D** depict the attenuation, shedding, and/or loss of the continuous ecGCx (arrows) with clumping of remaining LaN positive ecGCx (arrows) at ×4000, ×8000; 0.5 and 0.2 µm respectively in the BTBR *ob/ob* obese diabetic models. Furthermore, note the intact ACfp, Pcfp and intact Mt. Scale bars lower left corner of each panel and magnification in the upper right. CL = capillary neurovascular lumen; EC = endothelial cells.

**Figure 8 ijms-22-05427-f008:**
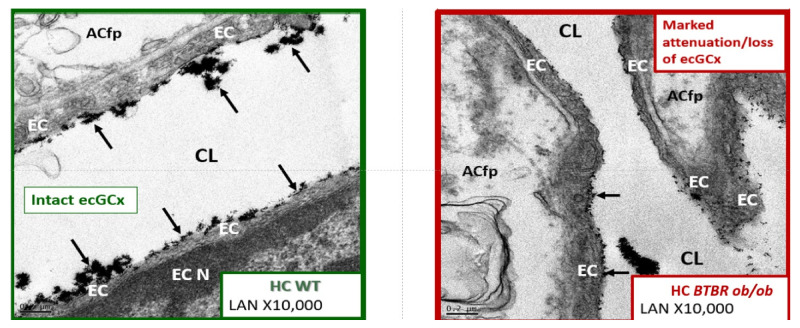
Lanthanum nitrate (LaN) staining of the endothelial glycocalyx (ecGCx) showing attenuation and loss in the obese diabetic BTBR *ob/ob* model: hippocampus CA-1 regions. Note the highly decorated endothelium by LaN staining in the control wild type (WT) hippocampus (HC) model with intact ecGCx (left-hand figure). In the obese diabetic BTBR *ob/ob* model, the ecGCx (arrows) when present is markedly attenuated and very thinned (right-hand figure). Magnification ×10,000; bar = 0.2 µm. ACfp = astrocyte foot process; CL = capillary lumen; EC = endothelial cell; HC = hippocampus; WT = wild type control.

**Figure 9 ijms-22-05427-f009:**
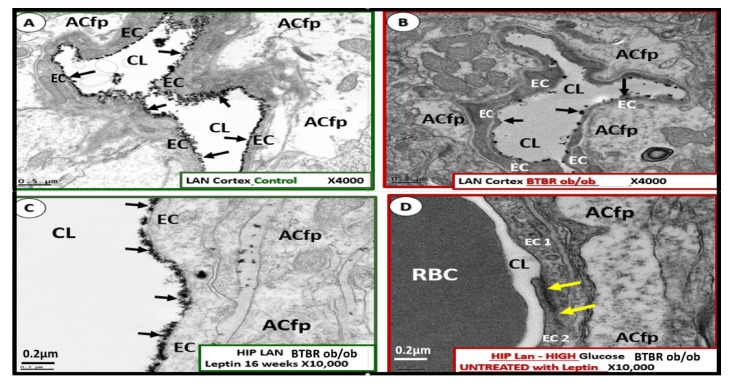
Leptin replacement in the obese diabetic BTBR *ob/ob* protects the brain endothelial cell glycocalyx (ecGCx) in cortical Layer III and hippocampus. **Panel A** illustrates the continuous decoration of the ecGCx with lanthanum nitrate (LaN) staining in the heterozygous non-diabetic control model cortical Layer III (arrows). **Panel B** depicts the marked attenuation and/or loss of the ecGCx in the obese diabetic BTBR *ob/ob* model cortical Layer III. Note when the ecGCx is present it is clumped and discontinuous (arrows). **Panel C** also illustrates the continuous decoration of the ecGCx in hippocampus CA-1 regions of the BTBR *ob/ob* models that were treated with intraperitoneal leptin for 16 weeks and stained with LaN (arrows). The ecGCx is continuous and comparable to the control model in **Panel A**. **Panel D** depicts the complete loss of the ecGCx by LaN staining in the hippocampus CA-1 regions of the BTBR *ob/ob*. Note that the tight and adherens junction (TJ/AJ) remain intact (yellow arrows). Magnification ×4000; scale bar = 0.5 μm in **Panels A** and **B**. Magnification ×10,000; scale bar = 0.2 µm in **Panels C** and **D**. ACfp = astrocyte foot process; Cl = capillary lumen; BEC = brain endothelial cell; HIP and HC = hippocampus CA-1 regions; LAN—LaN = lanthanum nitrate stained.

**Figure 10 ijms-22-05427-f010:**
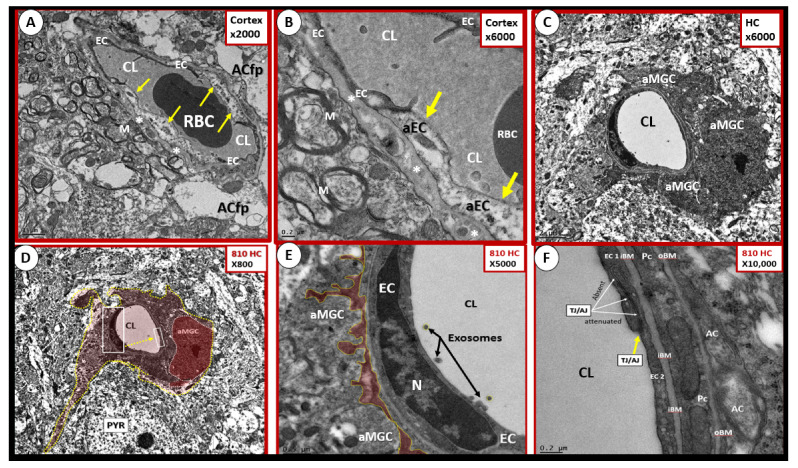
Endothelial and microglial activation in cortical and hippocampal NVUs of the BTBR *ob/ob* model. **Panels A** and **B** depict swollen electron lucent activated endothelial cells (aECs) and thickened basement membranes of cortical Layer III. Note the vacuolization of the BM in **Panels A** and **B** (asterisks). Magnification ×2000 and ×6000 respectively. **Panels C** through **F** depict an aMGC encroaching between the BEC and the rest of the NVU. **Panels C** and **D** depict an aMGC with loss of pericytes and astrocytes. **Panel D** (a lower magnification of **Panel C**) illustrates the aMGC pseudo-colored red and its relation to a pyramidal cell (PYR). **Panel E** depicts aMGC cytoplasmic processes (pseudo-colored red) directly abutting the ECs basement membrane and is a higher magnification of boxed in region in **Panel D**. Furthermore, note the exosomes within the capillary lumen that are between 50–75 nm in diameter, which may indicate stressed EC activation. **Panel F** illustrates that while these endothelial changes are occurring to the left and inferior NVU, the right or opposite side of the endothelial lining depicts the tight and adherens junction (TJ/AJ) remaining intact (arrows). Various magnifications labeled upper right and scale bars in each panel in lower left. ACfp = astrocyte foot processes; aMGC = activated microglial cell; CL = capillary lumen; iBM = inner basement membrane; M = myelin; n = nucleus; oBM = outer basement membrane; PYR = pyramidal cell; RBC = red blood cell. TJ/AJ = tight junctions/adherens junctions.

**Figure 11 ijms-22-05427-f011:**
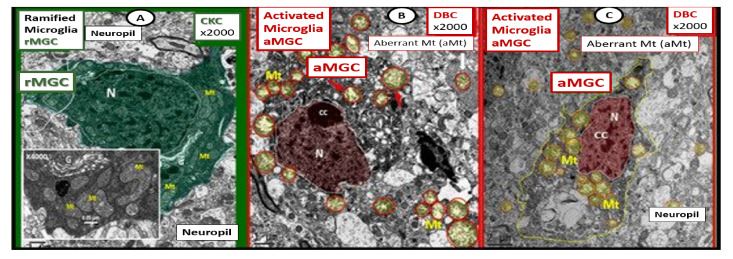
Comparison of ramified microglia (rMGC) in control mice to activated microglia (aMGC) in obese Type 2 diabetes mellitus (T2DM) *db/db* model. **Panel A** illustrates the normal appearance of ramified microglia (rMGC-pseudo-colored green) (see Figure 5A,C). In addition, note the insert in **Panel A** and the presence of cristae in rMGC at higher magnification ×6000). **Panels B** and **C** depict aMGCs (pseudo-colored red) with swollen electron lucent aberrant mitochondria (aMt) (pseudo-colored yellow) the *db/db* diabetic mice. Magnification ×2000; scale bar = 1 µm in **A** and **B** and 2 µm in **Panel C**. These images are adapted with permission from references [3,39]. CC = chromatin condensation; CKC = control C57BL/6J female non-diabetic model; DBC = female diabetic *db/db* model.

**Figure 12 ijms-22-05427-f012:**
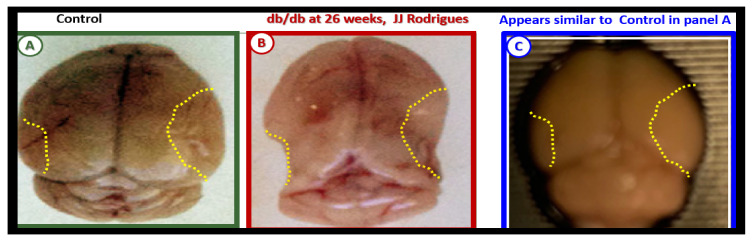
Gross brain atrophy in the *db/db* model but not in the diabetic BTBR *ob/ob* model. **Panel B** illustrates the brain atrophy at the time of surgical removal of the diabetic *db/db* models as compared to control models in **Panel A**. **Panel B** depicts the marked atrophy or loss in the cortical-parietal-hippocampal regions (outlined by the yellow dashed lines) in comparison to the control in **Panel A** at 26-weeks. **Panel C** demonstrates the BTBR *ob/ob* model at 20-weeks without atrophy in the parietal-hippocampal regions of the whole brain when compared to the *db/db* model in **Panel B** at 26-weeks. This may correlate with the less severe ultrastructural remodeling of the neurovascular unit and less microglial neuroinflammation in BTBR *ob/ob* mice as compared to *db/db* mice. **Panels A** and **B** CC by NC-ND [75].

**Figure 13 ijms-22-05427-f013:**
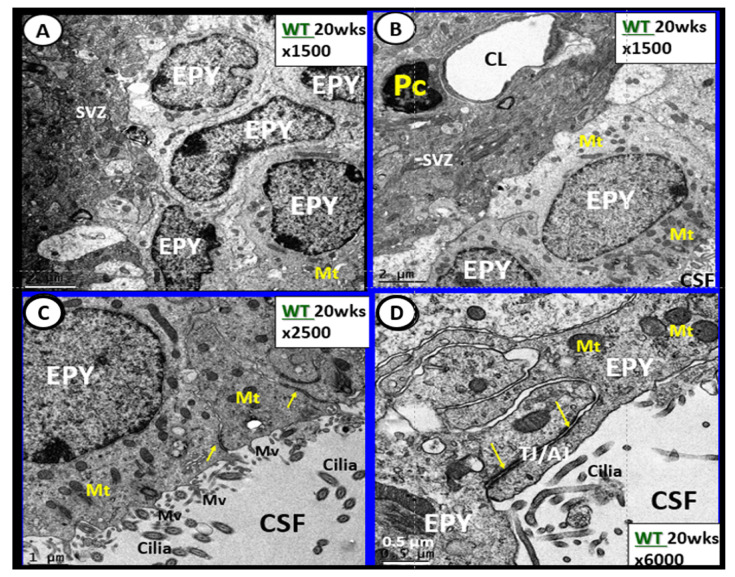
Ependymal cells lining the ventricular cerebrospinal fluid (CSF) system of the aqueduct in female control 20 week old models. **Panel A** illustrates that not all ependymal cells (EPY) are cuboidal but may lie flattened and be multilayered at the subventricular zone (SVZ) region of the adjacent neuropil. **Panel B** demonstrates the more classical cuboidal morphologic phenotype of the EPY cells that line the CSF ventricles. Note the close 5 µm distance between the EPY cells and fenestrated capillary within the SVZ neuropil. The adjacent capillary neurovascular units (NVU) in the SVZ do not have a similar coverage by astrocytes as in the cortical regions but pericytes (Pc) are frequently noted and this capillary appears to be fenestrated due the thinning of the ECs. **Panel C** depicts the ependymal cilia with the classical 9:2 arrangements of cytoskeletal proteins within them in addition to multiple microvilli (Mv) on the apical surface facing the CSF. **Panel D** demonstrates the overlapping and interdigitations of EPY cells, and also the staining of the very electron dense tight and adherens junction (TJ/AJ) proteins (arrows) and desmosomes that create the barrier functions of the EPY cells in some regions of the ependymal lining of the CSF. While these EPY cells are located in the aqueduct they also are representative of the EPY cells found throughout each of the four ventricles and lining of choroid plexus (CP) that provides the CP with its barrier function. As noted in these images, the EPY cells form the structural and functional barrier of the choroid plexus. CL = capillary lumen.

**Figure 14 ijms-22-05427-f014:**
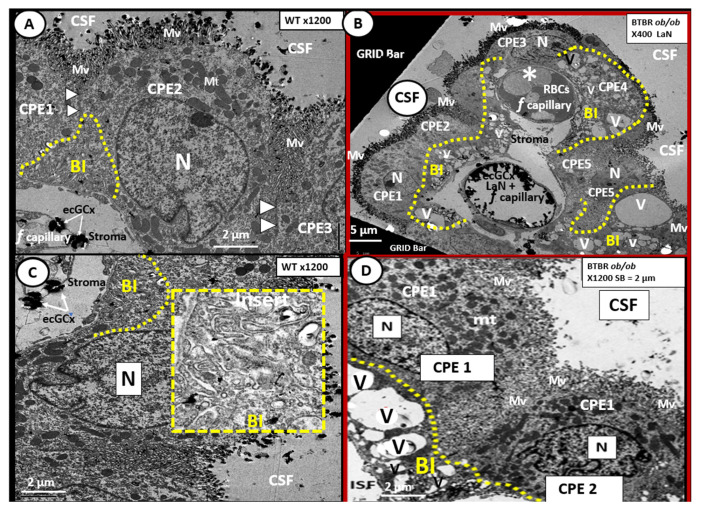
Choroid plexus (lateral ventricle) with aberrant vacuolization of the basilar infoldings of ependymal cells in obese diabetic female BTBR *ob/ob* models. **Panel A** demonstrates the normal ultrastructural morphology of the choroid plexus ependymal cell(s) (CPE) in the control heterozygous model, characterized by compact basilar infoldings (BI) (yellow dashed line), tight and adherens junctions (TJ/AJ) (arrowheads), microvilli (Mv) of the brush border at the apical cerebrospinal fluid (CSF) interface, and the multiple electron dense mitochondria (Mt) in a highly polarized ependymal cell. **Panel B** depicts aberrant remodeling changes of the (BI) at lower magnification to include multiple CPEs (CPE 1–5) which consist of vesiculation (v)/vacuolization (V) of the BI (yellow dashed lines). Importantly, note the lower fenestrated (*f*) capillary with positive staining for lanthanum nitrate (LaN) of the endothelial glycocalyx (ecCGx) and the upper capillary (*) that is filled with multiple red blood cells (RBCs) that is highly suggestive of capillary micro-thrombosis and without evidence of LaN staining. **Panel C** also illustrates the control heterozygote CPEs and this image contains an exploded insert of the BI (outlined in yellow dashed lines). This insert illustrates the compactness of the BIs without v/V. **Panel D** depicts the v/V of the BI as compared to the control models in **Panels A** and **C**. Magnification ×400; scale bar = 5 µm in **Panel B** and magnification ×1200; scale bar = 2 µm in **Panels A**–**D**. n = nucleus.

## Data Availability

Corresponding author will provide data upon reasonable request.

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
