# Peer review of "Deficient Leptin Cellular Signaling Plays a Key Role in Brain Ultrastructural Remodeling in Obesity and Type 2 Diabetes Mellitus"

_ijms, 2021, doi:10.3390/ijms22115427_

Round 1

Reviewer 1 Report

The authors have provided an incredibly detailed review examining the influential role of the hormone leptin on the ultrastructural remodelling which takes place in the cerebral space, specifically in the context disease states such as type 2 diabetes when leptin signalling is deficient. To begin, the authors detail the evolution of in vivo modelling of obesity/type 2 diabetes, before providing original data using cutting-edge models of leptin signalling deficiency to compare and contrast the findings of this model to those more established, all the while examining in great detail the changes witnessed across several cell types which comprise the neurovascular unit. Overall, this covers a topic which is quite pertinent in the area of type 2 diabetes/obesity research in the current day, and is not only a valuable resource for those interested in this area of research in terms of detailed background, but the original data also sheds light on one of the lesser covered areas of research within this field; brain physiology.

In reviewing the manuscript, I had however a number of concerns. The following should be addressed when preparing a suitable revision.  

  1. I would think that there are perhaps too many figures/details associated with the figures in this piece. For a review, 17 figures is an incredibly high amount, and the level of detail in the figures, in addition to the level of detail in the figure legends, is too much. This should be scaled back, with the majority of information delivered in the text.
  2. The figures themselves are too detailed/poor quality. The level of detail is perhaps too much, and most of the figures are overcrowded, often at the expense of the text within such. The authors should revise many of these figures, and improve the formatting, and in particular the resolution, with an aim of delivering clearer diagrams to complement the text.
  3. The figures should be referred to in the text. There are a few instances where figures stand alone and are not referenced in the text.
  4. If the authors intend to include original data, methods, even as a supplementary, should be included in some capacity.
  5. The piece is a little too self-referential at times. I appreciate this is the authors’ area of expertise, but as this is a review an effort should be made to reduce this, and instead be more encompassing of the area of research itself.
  6. The writing overall is very clear and detailed, however in some instances, some statements are vague and/or lack context. For example, at the start of Section 3, the opening statement is vague, and provides 8 references with no details given. Instances such as these should be reduced and addressed accordingly.  

Reviewer 2 Report

In my opinion, it may be hardly to understand the main theme of this review article. While the authors include 'leptin' in the title and explain about it at beginning, they mainly focus on the introduction of anatomical and molecular construction of neurovascular system in normal and type 2 diabetic animal model. The volume of information with figures is too much high as a review article, and therefore the theme should be specialized into neurovascular system in diabetic condition (with changing the title) and other information (leptin, diabetic model, etc.) should be simplified.

The background should be changed for showing summary of diabetes and influence of diabetes into neurovascular system.

The explanation of leptin should be summarized.

I think explanation of streptozotocin should be removed because of there is no importance in this review. By the way, streptozotocin-induced diabetic model is not equal to type 1 diabetes model (Figure 2).

Figures 4A and C are the same. No need to show both.

Figure 7B and C: information acquired from the figures is the same because of similar magnification (8,000 and 10,000). Should be summarized. Figures 7E and F also should be summarized because of the same reason.

Figures 13B and C are the same, should be summarized.

Figures should be reduced, summarized and well-designed.  

Author Response

Please see the attahcment

Round 2

Reviewer 1 Report

The authors have responded favourably to my comments, and as a result the manuscript is much improved. I would still have some reservations over the clarity of some details in some of the figures (some text is incredibly small), but this is a minor critique in what is a very detailed piece of work depicting the research area.   

Author Response

We thank the reviewer for the kind comments. In answering the 2nd reviewer comments, the number of panels have been reduced so that each panel in those figs will appear larger. We believe this will address the worst cases.

Reviewer 2 Report

The revised version becomes easier to understand because the theme becomes clear.

I recommended the revision about figures in previous review (Figures 4A and C are the same. No need to show both. Figure 7B and C: information acquired from the figures is the same because of similar magnification (8,000 and 10,000). Should be summarized. Figures 7E and F also should be summarized because of the same reason. Figures 13B and C are the same, should be summarized). You responded that the images referenced above in 4, 7, and 13 are either expansions or enlargements to guide the reader in seeing particular points in the image, however, I do not think this presentation contributes to the readers' understanding. Again, these figures should be summarized. For example, Figure 4A should be removed. Figure 7C and F should be removed. Figure 13C should be removed.

Author Response

We thank the reviewer for these helpful comments. We have removed the indicated duplicate panels.